# Increased variability in Greenland Ice Sheet runoff from satellite observations

Thomas Slater [1✉], Andrew Shepherd[1], Malcolm McMillan[2], Amber Leeson [2], Lin Gilbert [3], Alan Muir[3,4], Peter Kuipers Munneke[5], Brice Noël [5], Xavier Fettweis[6], Michiel van den Broeke [5] & Kate Briggs[1]

Runoff from the Greenland Ice Sheet has increased over recent decades affecting global sea level, regional ocean circulation, and coastal marine ecosystems, and it now accounts for most of the contemporary mass imbalance. Estimates of runoff are typically derived from regional climate models because satellite records have been limited to assessments of melting extent. Here, we use CryoSat-2 satellite altimetry to produce direct measurements of Greenland's runoff variability, based on seasonal changes in the ice sheet's surface elevation. Between 2011 and 2020, Greenland's ablation zone thinned on average by 1.4 ± 0.4 m each summer and thickened by 0.9 ± 0.4 m each winter. By adjusting for the steady-state divergence of ice, we estimate that runoff was 357 ± 58 Gt/yr on average – in close agreement with regional climate model simulations (root mean square difference of 47 to 60 Gt/yr). As well as being 21 % higher between 2011 and 2020 than over the preceding three decades, runoff is now also 60 % more variable from year-to-year as a consequence of large-scale fluctuations in atmospheric circulation. Because this variability is not captured in global climate model simulations, our satellite record of runoff should help to refine them and improve confidence in their projections.

---

[1] Centre for Polar Observation and Modelling, School of Earth and Environment, University of Leeds, Leeds, UK. [2] Lancaster Environment Centre, Lancaster University, Lancaster, UK. [3] Mullard Space Science Laboratory, Department of Space & Climate Physics, University College London, London, UK. [4] Centre for Polar Observation and Modelling, Department of Earth Sciences, University College London, London, UK. [5] Institute for Marine and Atmospheric research Utrecht, Utrecht University, Utrecht, the Netherlands. [6] SPHERES Research Unit, Department of Geography, University of Liège, Liège, Belgium.
✉email: t.slater1@leeds.ac.uk

 **1**

ce sheets are impacted by the atmosphere through a variety of surface mass balance (SMB) processes—primarily snowfall and runoff—and also lose mass to the oceans via glacier discharge. The Greenland Ice Sheet has lost mass over recent decades because ice discharge has exceeded its SMB[1,2], contributing an estimated $10.8 \pm 0.9$ mm to global sea levels since 1992[3]. Surface processes have been the principal driver of this imbalance because net SMB has declined as the regional climate has warmed[4]. In part, this is due to changes in atmospheric circulation[5] leading to increased runoff[3,6], behaviour which is likely to continue as global temperatures rise[7]. As well as raising global sea level, ice sheet runoff also delivers freshwater into the regional ocean[8], weakening deepwater convection[9] and reducing the strength of the Atlantic meridional overturning circulation[10]. Runoff also influences coastal marine ecosystem productivity in adjacent fjords[11] and sediment delivery to the North Atlantic and Arctic oceans[12]. En-route to the ocean, runoff also supplies water to surface[13], englacial[14] and subglacial[15] hydrological networks, which are known to influence rates of ice flow[16]. Although automatic weather stations provide point measurements of SMB components[17,18] and satellite observations are able to monitor trends in the extent of surface melting[19], regional climate models (RCMs, e.g.,[20–22]) have been the principal source of ice-sheet wide estimates of runoff.

Because Greenland SMB follows a strong seasonal cycle[23] with net ablation (runoff exceeding snowfall) and net accumulation (snowfall exceeding runoff) confined to summer and winter months, respectively, measurements of ice sheet elevation change offer an opportunity to monitor each process. Long-term signals of ice sheet surface lowering brought about by increases in runoff, firn densification and ice discharge have been resolved by satellite altimeters; in particular CryoSat-2[24–26] and ICESat-2[27] thanks to their fine spatial sampling. At the same time, ice sheet elevation changes arising from SMB and firn processes alone have also been reconstructed using RCMs and firn density models[28,29], and satellite altimetry and firn density models have been combined to partition ice sheet elevation changes into signals due to ice-dynamical and meteorological processes[3,25,30]. While RCMs have been able to capture recent interannual variability in runoff, global climate models (GCMs) often used in sea level projections have not, and have underestimated Greenland's contribution to sea-level rise as a consequence[31]. To date, however, the potential of satellite altimetry to directly monitor SMB has yet to be exploited. In this study, we use CryoSat-2 radar altimetry to separate long-term and seasonal elevation changes in the ablation zone of the Greenland Ice Sheet between 2011 and 2020, to directly measure runoff at the scale of the Greenland Ice Sheet and characterise its recent interannual variability.

## Results

### Ice sheet elevation change.
We use 51 million CryoSat-2 altimeter measurements acquired between January 2011 and October 2020 to compute surface elevation change across the Greenland Ice Sheet. From these data, we calculate linear trends in elevation (Fig. 1a) using a model fit method[25,26] applied within $5 \times 5$ km grid cells to separate elevation fluctuations evolving over time from those resulting from local topography and temporal variations in radar backscatter (see Methods). To evaluate the accuracy of the CryoSat-2 elevation trends, we compare them to 15,380 contemporaneous and independent estimates determined from airborne laser altimetry[32] and find a mean difference of 15 cm/yr. Thinning is concentrated at the ice sheet margin, consistent with the findings of previous surveys[25,26], likely as a result of increased surface melting[3,25,26] and the speedup of marine-terminating glaciers[33–35]. There is a notable signal of thickening at Storstrømmen in north-eastern Greenland, which is known to have slowed after surging

between 1978 and 1984[36]. We suggest a band of no elevation change just inland of the ice sheet margin in the southwest (Fig. 1a) is due to the formation of low-permeability ice slabs that have been identified in airborne laser altimetry and RCMs[37]; within this region, we find better agreement between airborne laser altimeter elevation trends and CryoSat-2 altimetry (mean difference of 3 cm/yr) than firn densification modelling ($-30$ cm/yr)[28]. In addition to elevation trends, we also compute the time-varying evolution of surface height (e.g., Fig. 2) by averaging monthly gridded elevation anomalies at 60-day intervals (see "Methods"). Time-varying elevation uncertainties are estimated as the average standard error of the aggregated elevation measurements, and we accumulate these over time in quadrature on the assumption that they are not temporally correlated.

We also simulate the ice sheet surface elevation change due to firn compaction and SMB processes using the Institute for Marine and Atmospheric Research Utrecht Firn Densification Model (IMAU-FDM)[28,29], forced by the RCM RACMO2.3p2[20] (see Methods). The IMAU-FDM output is restricted to the period January 2011 to December 2016, which provides 5 years of overlap with our CryoSat-2 data set. To allow the firn model to be compared to the satellite observations (e.g., Fig. 2), we resample its output to the spatial and temporal domain of our gridded CryoSat-2 data using bilinear interpolation; although the IMAU-FDM output is continuous in space, we subsample to locations common to the spatial coverage of the satellite observations at each epoch when making direct comparisons.

To examine elevation changes associated with different meteorological processes, we use RACMO2.3p2[20] simulations (which compare favourably to observations of the current climate[21]) to define the spatial extents of the ice sheet ablation, runoff and dry snow zones (Fig. 2). The ablation zone is taken to be the 211,225 km² area that falls below the equilibrium line (i.e. has negative SMB) in the majority of the years for which IMAU-FDM and CryoSat-2 data are available (between 2011 and 2017). The 'runoff zone' is taken as the 418,325 km² area for which runoff is predicted in the majority of years, and includes the ablation zone. The dry snow zone is taken as the 802,350 km² area for which no runoff emerges in the majority of years (although there has been melting on occasion[5]), and excludes the percolation zone where meltwater percolates and refreezes within the snowpack. On average, between 1980 and 2010, 80 % of annual total runoff emerges from the ablation zone and the rest arises from a 207,100 km² area just inland[20]. The ablation, runoff, and dry snow zones receive 12 %, 37 % and 27 % of the annual total snow accumulation, respectively[20].

### The seasonal cycle of melting and snowfall.
Over sub-annual timescales, ice sheet surface elevation changes in Greenland are dominated by the seasonal cycle of runoff in summer (May-August) and snowfall in winter (September–April). We take advantage of this distinct characteristic by computing the average seasonal rates of elevation change between 2011 and 2020 by fitting piecewise linear trends to the altimeter time-series in summer (Fig. 1b) and winter (Fig. 1c). For this calculation, we aggregated the altimeter data within coarser, $20 \times 20$ km grid cells to offset the reduced time period relative to the annual time series (see Methods). On average, the ablation zone thinned by $2.80 \pm 0.47$ m/yr in summer and thickened by $1.18 \pm 0.39$ m/yr in winter. Summer thinning increases toward the ice sheet margin and is largest and most widespread in the west and southwest, where surface temperatures are highest. Overall, summer thinning has far exceeded winter thickening across the ablation zone, which has experienced an average net thinning of $6.12 \pm 0.38$ m over this ten-year period as a result (Fig. 2). Across the ablation

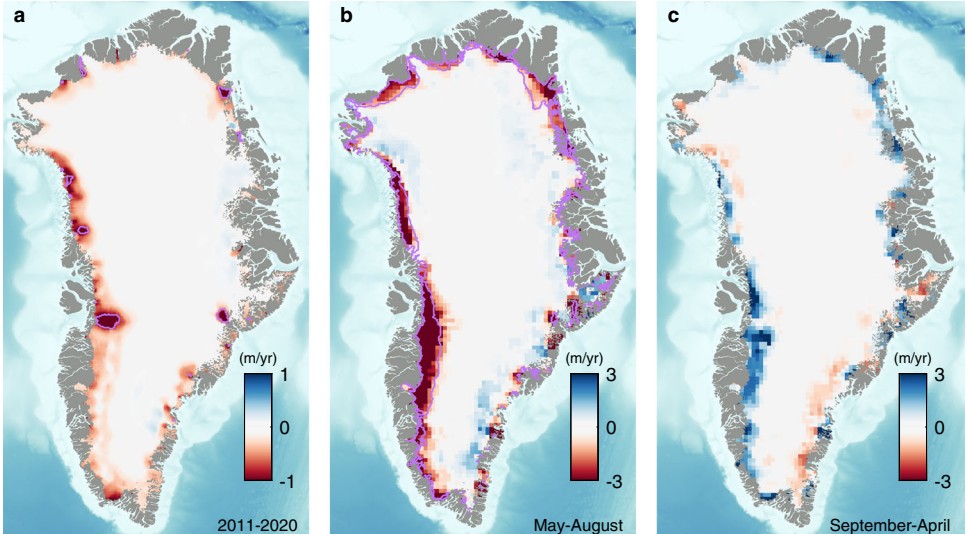

**Fig. 1 Greenland Ice Sheet interannual and seasonal elevation change from CryoSat-2 radar altimetry, 2011–2020. a** Rate of surface elevation change between 2011 and 2020. Purple contours depict areas of long-term dynamical imbalance, from repeat optical imagery[48,49] (1985–2018) and dynamic elevation trends determined from satellite altimetry and the Institute for Marine and Atmospheric Research Utrecht Firn Densification Model (2011–2017). **b** Average rates of elevation change during May–August and between 2011 and 2020. Purple contours depict the extent of the ice sheet ablation zone used in this study. **c** Average rates of elevation change during September–April between 2011 and 2020.

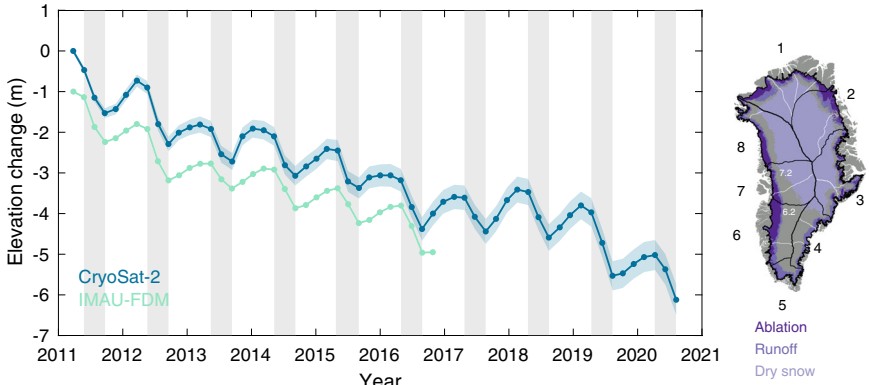

**Fig. 2 Surface height evolution in the ablation zone of the Greenland Ice Sheet, 2011–2020.** (left) Elevation change time-series derived from CryoSat-2 altimetry (dark blue, shaded region represents the estimated 1$\sigma$ uncertainty) and the Institute for Marine and Atmospheric Research Utrecht Firn Densification Model (IMAU-FDM) (light blue) for the ice sheet ablation zone between January 2011 and October 2020. IMAU-FDM time-series have been offset by 1 m for ease of viewing. (right) Definition of ice sheet facies and principal drainage basins (black) and sub-basins (white)[39] used in this study.

zone, where the density at which elevation fluctuations occur is well-constrained, the IMAU-FDM and CryoSat-2 elevation trends are in remarkably close agreement over both inter-annual and seasonal timescales (Fig. 2). During the period common to both datasets, the average per-epoch elevation changes across the ablation zone are strongly correlated in both summer ($R^2 = 0.98$) and winter ($R^2 = 0.96$), and agree to within (r.m.s. difference of) 23 cm overall (Fig. 3a). We also find good agreement between altimetry (39 cm) and IMAU-FDM (36 cm) height changes and airborne laser altimetry[38] in two drainage basins[39] where the airborne surveys are extensive enough to allow direct comparison (Fig. 3b) (see Methods).

Interannual variations in Greenland's ablation zone summertime elevation change from CryoSat-2 altimetry reflect variations in atmospheric forcing; net thinning was highest across the ice sheet ablation zone in the summers of 2012 (1.78 ± 0.24 m) and 2019 (1.91 ± 0.50 m), both record-high melt years. In 2012 and 2019 a strongly negative North Atlantic Oscillation (NAO) increased the prevalence of near-stationary high-pressure

systems[5], favouring warm air advection and sunny conditions which enhance melt-albedo feedbacks[40]. In both summers, warm air was advected over much of the ice sheet margin, leading to increased surface melting. By contrast, summer thinning was 1.21 ± 0.41 m on average between 2013 and 2015 — 45 % lower than in 2019 — when atmospheric circulation brought about low-temperature and more cloudy conditions, inhibiting melting[41]. During our survey period, elevation gains in winter due to snowfall were 50 % lower in magnitude on average than thinning in summer, reaching a maximum of 1.22 ± 0.44 m between September 2017 and April 2018 when snowfall was high across the entire ice sheet[3,40].

To explore the seasonal elevation changes in greater detail, we also examined trends within 7 of the ice sheets 8 principal drainage sectors[39]. We exclude the southeast (Basin 4) from our overall statistics, because the ablation zone is narrow (< 1 % of its total extent) and because the sector has a rugged terrain that is challenging for both altimetry and regional climate modelling. At the basin scale, seasonal elevation changes from CryoSat-2

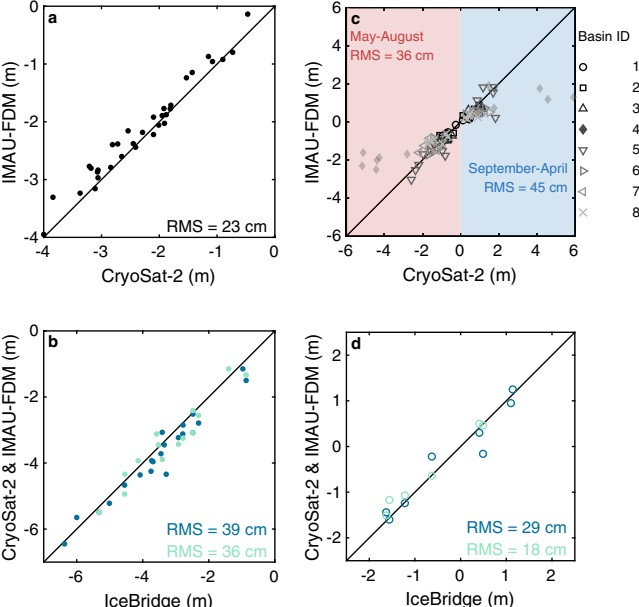

**Fig. 3 Surface height evolution and seasonal elevation changes in the ablation zone of the Greenland Ice Sheet from CryoSat-2 altimetry, firn modelling and airborne laser altimetry. a** Institute for Marine and Atmospheric Research Utrecht Firn Densification Model (IMAU-FDM) versus CryoSat-2 60-day estimates of height evolution in the ice sheet ablation zone between 2011 and 2016. **b** CryoSat-2 (dark blue) and IMAU-FDM (light blue) height evolution versus Operation IceBridge laser altimeter measurements[38] between 2011 and 2018 acquired in sectors 6.2 and 7.2 (Fig. 2). **c** IMAU-FDM versus CryoSat-2 summer and winter elevation changes in the ice sheet ablation zone between 2011 and 2017 and in 7 principal ice sheet drainage basins[39]. For completeness, we include Basin 4 here, but exclude it from our reported statistics, because the ablation zone there is narrow (<1 % of its total extent), and has a rugged terrain that is challenging for both altimetry and regional climate modelling. **d** CryoSat-2 (dark blue) and IMAU-FDM (light blue) versus Operation IceBridge seasonal elevation changes between 2015 and 2018 in sectors 6.2 and 7.2 (Fig. 2). In each panel, the root mean square difference (RMS) between the respective datasets is provided.

altimetry agree with those derived from IMAU-FDM to within 45 cm in winter and 36 cm in summer (Fig. 3c) (including Basin 4, these rise to 109 cm in winter and 94 cm in summer). These differences are comparable to the estimated uncertainty of both the altimeter data (50 cm) and IMAU-FDM (20 to 40 cm/yr)[28]. However, we note that CryoSat-2 agrees with airborne laser altimeter estimates to be within 29 cm (Fig. 3d). At the regional scale as well as the ice sheet scale, CryoSat-2 captures the spatial and temporal variability expected from variations in atmospheric forcing: thinning associated with the extreme summer of 2012 is highest in the western (basin 7) and southwestern (basin 6) sectors (2.19 ± 0.35 m and 1.68 ± 0.12 m on average, respectively), over which warm southerly air has been advected during a negative NAO phase[5] and where early inland snowline migration reduced surface albedo and enhanced melt[20,42]. Surface lowering significantly increased in all sectors during the summer of 2019, reaching a maximum of 2.77 ± 0.31 m in the northwest when warm air was advected towards the north and cloud cover was below average in the south[40].

The close agreement between interannual and seasonal changes in elevation recorded by CryoSat-2 and the IMAU-FDM (Fig. 3) suggest that SMB processes are the primary driver. Residual differences will arise due to signals of ice dynamical imbalance that are not considered by the firn model, or due to uncertainty in

the individual datasets. While there are many examples of long-term ice dynamical imbalance across the Greenland Ice Sheet[33,34,43], seasonal variations in ice flow are considerably smaller in magnitude[44], and the area of ice affected by them is, in any case, a minor fraction of the ablation zone. Indeed, when areas of ice dynamical imbalance are excluded, there is little change in either the seasonal elevation changes or their difference relative to the IMAU-FDM (see Methods). Potential sources of error in the IMAU-FDM include, but are not limited to, an inadequate choice of reference period selected for initialisation, or the reduced ability of the RCM forcing data to sufficiently capture changes in ice albedo and sensible heat flux in low-lying regions[20,28,45]. Satellite altimetry also struggles when monitoring rugged terrain, which is particularly apparent in the eastern and southern ice sheet sectors (basins 3, 4 and 5); the same is true for RCMs when resolving steep slopes[46]. Seasonal changes in the scattering properties of the firn layer could also lead to a poorer agreement between the satellite altimetry and IMAU-FDM, especially in the southeast sector where spurious signals associated with tracking of submerging ice lenses have been detected[47], and there is some evidence of these effects in our elevation trends in the south-east inland from the ablation zone (Fig. 1). However, fluctuations in firn penetration should not affect long-term elevation trends recorded in the ablation zone itself, because the winter snow layer melts down to bare ice each year.

**An observational record of ice sheet runoff.** Because seasonal changes in ice sheet elevation are primarily driven by SMB, and because the dominant process during summer is ice melting[6,20], we use our elevation trends as the origin of an observationally-based estimate of runoff. To do this, we first identify and exclude areas of long-term ice dynamical imbalance from the altimeter record with the aid of ice velocity data[48,49] (see Methods), because their elevation changes are not solely due to SMB. These areas represent just 5 % (20,525 km²) of the runoff zone (Fig. 1a) and coincide with fast-flowing marine-terminating glaciers that are known to be in a state of imbalance[33,34,43,44,50,51], including areas that exhibit seasonal changes (e.g.,[44,52,53]) as well as dynamic thinning of up to 2 m/yr. Across the remaining area, we adjust the altimeter summer height changes within each 5 × 5 km grid cell to remove the elevation signal associated with the steady-state divergence of ice and thereby isolate the contribution due to SMB anomalies alone (see Methods). We then use the resulting elevation change anomalies to estimate summertime changes in mass by aggregating them across two regions and applying fixed densities.

In the ablation zone, we assume that elevation change anomalies occur at the density of ice (917 kg/m³), and across the inland runoff zone, and when the elevation changes are negative, we assume they occur at the density of firn. We obtained a firn density of 684 kg/m³ from shallow cores collected recently in the region[54–56] rather than ice-sheet wide models as they are known to be biased high in the inland runoff zone[56], though the choice has only a small (< 10 %) impact on our runoff solution because the region makes a small overall contribution (see Methods). Our approach is a simplification because it neglects mass accumulated during the summer period. However, although this simplification could in principle lead to an underestimate of runoff, our choice of ice and firn densities mitigate this because they represent upper and contemporary bounds, respectively, and in any case, the simplification has only a minor impact because snowfall is a small fraction (15 %) of the summertime SMB across the runoff zone (see Methods). To verify this, we compared model estimates of SMB and runoff within the ablation and runoff

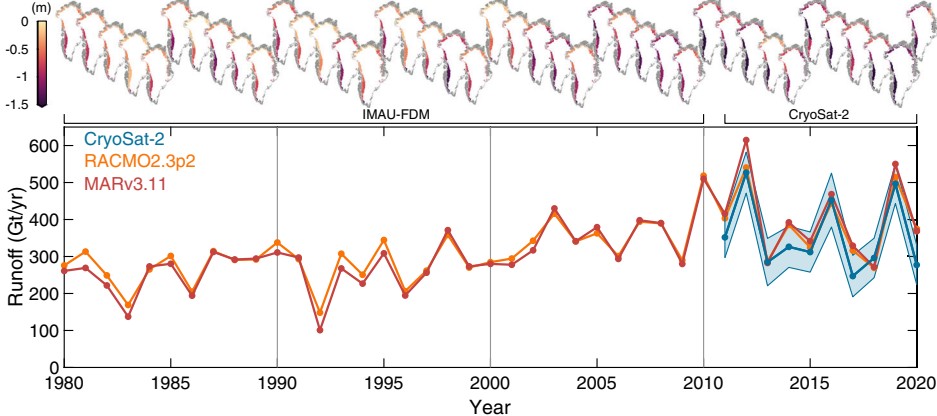

**Fig. 4 Greenland Ice Sheet runoff and summer elevation change, 1980–2020.** (bottom) Runoff from the Greenland Ice Sheet derived from RACMO2.3p2[20] (orange, 1980–2019) and MARv3.11[22] (red, 1980–2020) regional climate models, and CryoSat-2 satellite observations (blue, 2011–2020, shaded region represents the estimated $1\sigma$ uncertainty). Vertical lines mark consecutive decades during the 40-year period (Table 1). (top) May–August elevation changes in the ablation zone of the Greenland Ice Sheet, divided into its principal drainage basins[39], derived from the Institute for Marine and Atmospheric Research Utrecht Firn Densification Model (IMAU-FDM) (1980–2010) and CryoSat-2 satellite observations (2011–2020).

**Table 1 Greenland runoff variability. The standard deviation of Greenland Ice Sheet runoff (in Gt/yr) per decade between 1980 and 2020.**

|  | 1980s | 1990s | 2000s | 2010s[a] |
|---|---|---|---|---|
| CryoSat-2 | – | – | – | 99 |
| MARv3.11 | 54 | 74 | 57 | 111 |
| RACMO 2.3p2 | 48 | 66 | 48 | 101 |

[a]2010s: this decade is not entirely covered by CryoSat-2 (2011–2020).

zones, and find them to be highly correlated ($R^2 = 0.99$ and 0.85, respectively)[20] — indicating that other processes play a minor role.

Our satellite observations show that runoff from the Greenland Ice Sheet was $357 \pm 58$ Gt/yr on average between 2011 and 2020 (Fig. 4). During this period, year-to-year variability in runoff was high, with a maximum spread of 280 Gt/yr and a standard deviation of 99 Gt/yr (Table 1). As a consequence of an exceptionally warm summer, runoff peaked at $527 \pm 56$ Gt/yr in 2012, reducing to $285 \pm 64$ Gt/yr in the following year when the NAO shifted abruptly to a positive phase and atmospheric conditions were cooler[41]. We record a runoff of $496 \pm 53$ Gt/yr in 2019, a year when the persistence of anticyclonic conditions increased air temperatures and drove record surface melting in the north[40]. Runoff reduced again in 2020 to $277 \pm 53$ Gt/yr, during a runoff season punctuated by heavy precipitation which increased albedo and reduced melt, and was similar to the long-term (1980–2010) average simulated by RCMs (296 Gt/yr)[20]. Our record shows interannual variability in Greenland runoff between 2011 and 2020 has been 60 % higher than in the previous three decades (Table 1), and twice as variable than the preceding decade (2000–2010), in response to recent large-scale fluctuations in atmospheric circulation over the ice sheet.

## Discussion
The CryoSat-2 based estimates of runoff are in close agreement with those derived from both the RACMO2.3p2[20] and MARv3.11 RCMs[22] (Fig. 4), which provide the current best-modelled estimates of Greenland Ice Sheet runoff[21]. Across the ablation zone, the root mean square difference between CryoSat-2 based estimates are 47 and 60 Gt/yr for RACMO2.3p2 and MARv3.11, respectively — comparable to the estimated uncertainty — over

their common 10-year period. Both modelled and CryoSat-2 estimates reflect the recent interannual variability in runoff and are well correlated ($R^2 = 0.85$ and 0.88 for RACMO2.3p2 and MARv3.11, respectively). We find the largest relative departure from both models in 2020, when runoff recorded by CryoSat-2 is 97 and 90 Gt/yr lower than RACMO2.3p2 and MARv3.11, respectively, but still agree within their respective uncertainties (which is estimated to be 15–20% for the RCMs[20,21,57]) for this and all years surveyed by CryoSat-2. Over the longer term, RCMs[20,22] show that there has been a progressive increase in summertime melting and thinning across the ice sheet ablation zone (e.g., Fig. 4). In the southwest (basin 7), for example, the rate of thinning has doubled from 0.76 m during the 1980s to 1.54 m on average during the period of our satellite altimeter survey (2011–2020), and runoff from the ice sheet overall was 34 % larger. Prior to 2010, runoff had never exceeded 500 Gt/yr in either model, and the highs we have recorded in 2012 and 2019 are 78 % and 68 % higher than the long-term (1980–2010) average (296 Gt/yr)[20].

Our method provides a record of seasonal elevation changes across the Greenland Ice Sheet, and the first observational and satellite-based estimate of ice sheet runoff at scale. Between 2011 and 2020, the ablation zone thinned by $0.5 \pm 0.1$ m/yr, on average, because summer thinning was more than twice as large as winter thickening. In total, the ice sheet lost $3571 \pm 182$ Gt of ice through runoff over the 10-year survey period, with record-breaking losses of $527 \pm 56$ Gt/yr first in 2012 and then $496 \pm 53$ Gt/yr in 2019. These highs occurred in summers when atmospheric conditions promoted strong runoff from the ablation zone, in concert with widespread melting across much of the ice sheet surface[5,40]. In comparison to the long-term (1980–2010) mean[20], runoff during the period 2011–2020 has been 21% higher, on average. However, the past decade has also seen a marked increase in interannual runoff variability, with a standard deviation of 99 Gt/yr — 60% higher than the previous three decades as determined from RCMs, and twice as high as the preceding one — and a minimum of $247 \pm 56$ Gt/yr in 2017 when the melt season was short in comparison and snowfall in June reduced melt-albedo feedbacks. This suggests that the freshwater outflow and ice discharge from the ice sheet may become more variable as the climate warms, as well as more intense, which may have implications for ocean circulation[10] and ecosystem productivity[58]. While not captured in the GCMs used in sea-level projections to date, this variability should be accounted for to accurately predict a sea-level rise in

the future[31]; our observational-based approach can help improve descriptions of the ice sheet–atmosphere interactions in climate models and improve confidence in their projections.

Because our approach is based on satellite observations, it allows runoff to be measured across the entire ice sheet and in near real-time and, potentially, backwards in time using measurements acquired by older satellite altimeter missions. It also provides an independent method for monitoring SMB processes based upon satellite observations, which can support improvements in RCM and GCM capability[17,57], general improvements in SMB models[21], and assessments of climate projection skill[59]. Our method demonstrates the exciting potential of satellite altimetry to monitor ice sheet SMB processes, and the next step is to extend our method to include an assessment of snowfall variability.

## Methods

**Computing elevation and elevation change**. We compute linear rates of surface elevation change (Fig. 1) and 60-day height evolution time-series (Fig. 2) from CryoSat-2 radar altimeter observations acquired between January 2011 and October 2020. In total, we used over 51 million measurements of ice sheet elevation provided in the baseline-D level 2i product, which are corrected for echo deviations from the on-board tracking gate, off-nadir ranging due to slope, dry atmosphere, wet atmosphere, ionosphere and solid-earth tide[60]. Using a model fit method (e.g.,[25,61,62]) we generate time-series and trends of ice sheet surface elevation change on a 5 × 5 km grid, allowing for elevation fluctuations within each grid cell caused by topography $(x, y)$, satellite heading $(h)$ and time $(t)$

$$z(x, y, t, h) = \bar{z} + a_0 x + a_1 y + a_2 x^2 + a_3 y^2 + a_4 xy + a_5 h + a_6 t \quad (1)$$

We solve for the individual model coefficients using an iterative least-squares fit to minimise the impact of outliers and discard any unrealistic estimates from poorly constrained solutions based on a set of statistical thresholds which include: a minimum of 40 data points, a time-series length of at least 2 years, a maximum root mean squared difference of elevation residuals from the model of 12 m, a maximum elevation rate magnitude of 10 m/yr, and a maximum surface slope of 5°.

To account for temporal variations in range associated with changes in radar echo shape, we apply an empirical correction based upon correlated changes in elevation and backscattered power[63,64]. We first compute the correlation gradient in elevation as a function of power, $\frac{dz}{dp}$, using a linear fit in each grid cell over a 60-month time window. We then multiply the time-series of changes in backscattered power $dp$ by this gradient to estimate the backscatter correction term, which we remove from our original elevation change time-series $dh$

$$dz_{corrected} = dz - \left( dp \frac{dz}{dp} \right) \quad (2)$$

Although previous studies have also adapted backscatter corrections to account for the effects of an episodic change in snowpack characteristics on the shape of altimeter echoes[25,62] due to widespread melting in the ice sheet interior[65,66], here we do not, as (i) the threshold offset centre of gravity retracker we use[67] is less sensitive to changes in volume scatter than physically based algorithms, and (ii) we only examine elevation trends in the ice sheet interior, which are less affected by the melt event when determined over longer time periods[68]. At the ice sheet margins where CryoSat-2 operates in SARIn mode, we use the ESA Level 2 SARIn retracker, which fits an analytical model to individual SAR waveforms[69].

Across a small proportion (9 %) of the ice sheet area our method fails to retrieve a solution, and we estimate elevation trends in this region instead of using an empirical model based upon latitude $(l)$, elevation $(z)$ and ice velocity $(v)$, all of which affect surface elevation change through temperature-related processes or ice flow[25]

$$\frac{dz}{dt} = al + bz + cv + d \quad (3)$$

We use annual surface velocity measurements acquired in 2017 from Synthetic Aperture Radar[70]. In this year, regular 6-day coverage provided by the Sentinel-1A and -1B satellites improved performance on fast-moving glaciers. Model coefficients are computed based on a least-squares fit to the available elevation change measurements. Where no velocity data are available, we model elevation changes as a function of latitude and elevation only. Based upon the root mean squared difference of the residuals from the model fit, we estimate an average uncertainty in unobserved grid cells of 0.4 m/yr.

To evaluate our elevation change rates derived from CryoSat-2 altimetry (Fig. 1), we compare our results to 15,380 contemporaneous and independent elevation trends derived from Operation IceBridge airborne laser altimetry[32] (Supplementary Fig. 1) acquired between 2011 and 2018. We remove any elevation rates for which the repeat period is less than 2 years or with a magnitude greater than 10 m/yr. We bin the IceBridge measurements at the same resolution as the

CryoSat-2 trends (5 km), and remove any grid cells sampled by less than 10 measurements or where the standard deviation is greater than 2 m/yr. We find the mean and standard deviation of the differences (CryoSat—IceBridge) of 15, and 64 cm/yr, respectively.

We compute time-varying surface height evolution by averaging gridded elevation anomalies using Eq. (1) and corrected with Eq. (2) (Fig. 2, Supplementary Fig. 2) within 60-day intervals. We quantify the uncertainty on the altimeter elevation estimates at each epoch by computing the regional average of the standard error of height change within all contributing pixels. Assuming this component is temporally uncorrelated, we then accumulate all preceding uncertainties in quadrature at any given epoch[71].

**Computing seasonal elevation trends**. We define summer as May 1st to August 31st, with winter defined as the surrounding months. To derive seasonal elevation changes in the ablation zone, we smooth the elevation time-series using a 180-day Gaussian-weighted moving average and take the peak to trough difference within each summer and vice versa in each winter period. To define the uncertainty on the seasonal height change, we sum the uncertainties associated with the two elevation estimates used to compute the seasonal change in quadrature. We find little difference between seasonal elevation changes computed when areas of long-term dynamic imbalance (Fig. 1) are removed (Supplementary Fig. 3): we find a root mean squared difference of 14 cm between seasonal elevation changes computed from CryoSat-2 where areas of dynamic imbalance are identified (basins 1–4, 7, 8, see Computing runoff, below), and a root mean squared difference of 36 and 45 cm between seasonal elevation changes computed from IMAU-FDM between May–August and September–March, respectively (see Simulating elevation change due to SMB processes, below).

To compute the average rate of seasonal elevation change from Cryosat-2 data (Fig. 1) between 2011 and 2020, we first aggregate our 5 × 5 km time series onto a coarser 20 × 20 km grid to reduce the impact of local topography on the resulting trends. For each 20 × 20 km grid cell and for each year between 2011 and 2020, we align elevation change time-series in successive seasons relative to the central value of the first summer or winter. We then fit a linear trend over the combined seasonal time series to determine the average seasonal elevation rate. To obtain the error on the seasonal rate of elevation change, we combine the elevation uncertainty with the standard error of the linear seasonal surface elevation trend in quadrature, in order to account for systematic errors that may affect the trend.

**Computing runoff**. In some places, changes in elevation occur due to changes in ice flow as well as SMB. To exclude these regions from our runoff calculation, we identify areas of long-term dynamical imbalance using a combination of ice sheet surface velocities determined from repeat optical imagery[48,49] (1985–2018) and dynamic elevation trends determined from satellite altimetry and IMAU-FDM (2011–2017). To identify areas of long-term dynamical imbalance from the ice velocity data, we fit linear trends to time series grouped within 240 m pixels which are flowing faster than 50 m/yr and contain measurements on at least 10 occasions. From these trends, we define areas where changes in ice speed are significant in both their magnitude (absolute acceleration greater than or equal to 5 m/yr²) and in their variability (standard deviation of the trend greater than or equal to 1 m/yr²) to be in a state of dynamical imbalance (Supplementary Fig. 4). These areas have sped up by 275 m/yr on average, since 1985. Because the ice velocity data do not provide complete repeat coverage of the ice sheet, we augment them with areas of dynamical imbalance determined from our CryoSat-2 elevation trends. To do this, we estimate contemporary elevation trends associated with dynamic processes by removing the signal due to surface processes alone using the IMAU-FDM. We then identify regions of dynamical imbalance as all locations where the dynamic elevation trend is greater than the average rate (0.4 m/yr) determined in areas of dynamical imbalance identified in the ice velocity data (Supplementary Fig. 4). In practice, our solution is relatively insensitive to the rate selected to mark areas of dynamical imbalance, varying by only 36 Gt/yr for rates between 0.2 and 1.0 m/yr (Supplementary Table 1). We then combine areas of dynamical imbalance identified in the ice velocity and elevation data by adding them together, apply a sequence of morphological closing operations (dilation followed by erosion), and removing areas not connected to the grounding line (Fig. 1). In total, we identified an area of 20,525 km² to be in a state of dynamical imbalance, 5 % of the total runoff zone area.

Because runoff occurs even when the ice sheet is in steady-state, in order to compute the absolute runoff from CryoSat-2 elevation measurements (Fig. 4) we apply a height correction to observed summer height changes within 5 × 5 km grid cells to account for the surface height change associated with the divergence of ice. For an ice sheet in steady-state, the change in ice thickness due to surface processes in a vertical column of ice is balanced by the ice flux through it; we estimate this height change by taking the long-term SMB mean in the ablation zone from the RACMO 2.3p2 RCM[20] during a period when the ice sheet was considered to be in a state of balance (1960–1980). In this way, we assume that (1) in the runoff zone and away from areas of dynamical imbalance, thickness changes due to ice divergence are approximately equal to the long term SMB mean and (2) because the dynamic response to SMB changes occurs over multi-decadal timescales[72], present-day observed departures in thickness in these areas are dominated by changes in SMB. In the absence of direct observations, we assess the validity of this

assumption by comparing dynamic and SMB-related ice sheet thickness changes derived from an ensemble of ISMIP6 model experiments, forced by a high-end emission scenario (RCP8.5)[72] (Supplementary Fig. 5, Supplementary Table 2). Within the area we calculate runoff from CryoSat-2 elevation measurements, modelled rates of elevation change due to SMB processes (−0.86 m/yr) are almost 30 times higher than those due to ice dynamics (−0.03 m/yr), even by the year 2100 when the ice sheet has been out of balance for a century[72] (Supplementary Table 2). These projections, along with observations which indicate that the ice sheet was approximately in balance pre-2000[1,3], suggest that contemporary changes in ice thickness reflect SMB anomalies, not divergence anomalies and that our assumption that thickness changes due to steady-state ice divergence are approximately equal to the long-term SMB mean is valid.

Taking the fraction of the height change due to advection ($dz$) which occurs in the summer, we apply a height correction in each grid cell which adjusts the observed summer surface height change due to surface processes ($z_f$), assumed to be the total summer height change according to CryoSat-2 within each grid cell, see Seasonal elevation changes), in order to estimate the height change due to runoff ($z_r$) (Supplementary Fig. 6)

$$z_r = z_f - \left(\frac{t_{summer}}{12}\right) dz \qquad (4)$$

Where $t_{summer}$ is the length of the summer period in months. On average this correction adjusts the summer height change by 11 cm and 2 cm in the ablation and entire runoff zones, respectively, reaching 50 cm in grid cells towards the ice sheet margins where steady-state melting is highest (Supplementary Fig. 6).

In the runoff zone, and outside areas of dynamical imbalance, we assume the observed elevation changes represent the surface height change due to all SMB processes integrated over the summer, and that during our chosen summer period the SMB is dominated by runoff. Within the ablation and inland runoff zones, we find that modelled summer SMB and annual runoff are highly correlated ($R^2 = 0.99$ and 0.85, respectively)[20], lending confidence to this assumption. Nevertheless, this is a simplification because our solution does not account for the effects of summertime snowfall, sublimation and drifting snow erosion on observed surface height changes. Sublimation and windblown divergence represent limited contributions to SMB (10 Gt/yr and less than 1 Gt/yr in the runoff zone, respectively)[20]. During our observational period, summertime snowfall amounts to 22 and 32 Gt within the ablation and inland runoff zones on average; 7 % and 33 % of the modelled runoff, respectively, and 13 % of the total runoff[20]. The effects of this on our solution will be dependent on the snowfall conditions within a given year: the melting of a thick surface snow layer at the start of the runoff season, present due to heavy snowfall in the preceding winter, for example, could bias our runoff estimates high due to our choice of higher densities in both the ablation and inland runoff zones. We note that in the inland runoff zone, where snowfall is higher, our choice of a lower density acts to reduce this bias in part, and that modelled surface densities typically vary by 14 and 9 kg/m³ at the beginning of May in the ablation and inland runoff zones, respectively[28]. Conversely, any snowfall in the summer which runs off the ice sheet will not be captured by our observations, and our solution could underestimate the total runoff. Taking both of these effects into account, snowfall equal to the modelled average will bias our runoff by 5 % and 21 % within the ablation and inland runoff zones, respectively, and 8 % overall; however these effects may be larger for years with heavy snowfall.

In the ablation zone, we convert the resulting volume changes in each grid cell to mass at a density of 917 kg/m³. In the remaining area of the runoff zone inland from the ablation zone, and during summer epochs when the volume change recorded by CryoSat-2 in the region is negative, we select a density of 684 kg/m³, based upon the average density recorded in the top 5 m of 5 firn cores obtained in the region[54–56] (Supplementary Table 3). We prefer to use firn cores to inform this choice of density as firn models have been shown to overestimate densities by 21 % at sites below 2000 m altitude[56]. We then sum the runoff from the two regions to calculate the total ice sheet runoff. To test the sensitivity of our runoff solution to our choice of density in the inland region, we compute total ice sheet runoff using a range of densities between 450 and 917 kg/m³ (Supplementary Fig. 8, Supplementary Table 4). We find total ice sheet runoff varies by 34 Gt/yr, 10 % of the average runoff computed using our chosen density and smaller than the estimated uncertainty (58 Gt/yr).

Because CryoSat-2 does not fully sample the ablation zone during a given 60-day period, we scale our mass change estimates at each epoch according to the proportion of the observed area within 500 m elevation bands (defined using the Greenland Mapping Project DEM[73]), before aggregating across the ablation zone and within each calendar year to estimate the annual runoff. As with our estimates of elevation change, we quantify the uncertainty in the altimeter runoff estimates at each epoch by computing the regional average of the standard error of the mass change within all contributing pixels and assume there is no error associated with our choice of density. To estimate the uncertainty in the annual runoff, we sum the uncertainties from each summer epoch within each calendar year in quadrature.

**Simulating elevation change due to SMB processes.** We simulate surface elevation changes due to firn compaction and SMB processes during the period of our satellite altimeter record using the Institute for Marine and Atmospheric Research

Utrecht Firn Densification Model (IMAU-FDM)[29]. IMAU-FDM simulates the time evolution of firn compaction, temperature, liquid water content, and surface elevation in a vertical 1-D column of firn and ice at high spatial (11 km) and temporal (10 days) resolution[28,29,74]. IMAU-FDM uses an expression for firn-densification[75] adapted to fit in situ density profiles retrieved from the Greenland Ice Sheet[28] and covers the period 1960–2017. The surface layer of the firn column is forced by SMB components (solid and liquid precipitation, surface and drifting snow sublimation, drifting snow erosion, surface melt), surface temperature and 10 m wind speed at 11 km horizontal and sub-daily (3–6 h) temporal resolution from the RACMO2.3p2 RCM[20]. Surface elevation changes are computed with respect to a spin-up period (1960–1979), over which zero surface elevation change due to firn and SMB processes is assumed[28]. In the ablation zone, where there is no persistent firn layer, elevation changes are modelled due to SMB processes alone.

To compare firn model outputs to the satellite observations, we resample the gridded firn model time-series to a spatial (5 × 5 km) and temporal (60-day) domain common to that used for the radar altimetry through bilinear interpolation. We then obtain average rates of elevation change due to SMB and firn processes through fitting a linear trend to the cumulative firn height anomaly time series in each grid cell. In a similar manner to the Cryosat-2 data, we compute time-varying surface height evolution in the ice sheet ablation zone at 60-day intervals by spatially averaging gridded monthly elevation anomalies.

**Comparing satellite and airborne elevation changes.** We compared time series of height evolution and seasonal elevation changes from both satellite altimetry and IMAU-FDM to independent estimates derived from airborne laser altimeter measurements[38] (Fig. 3). We perform the comparison using data acquired between 2011 and 2019 in two drainage basins[39] where the airborne surveys are extensive enough to allow averages to be representative of the whole area (Supplementary Fig. 9). We derive time series of height change from airborne laser altimetry within each region by averaging the height difference between measurements co-located within a radius of 10 m. Flight lines are repeated in the spring of each year, as well as in the autumns of 2015, 2016 and 2019, allowing us to compare seasonal elevation changes as well as interannual changes (Fig. 3). We note that, within the selected regions, differences between elevation changes recorded by CryoSat-2, IMAU-FDM and airborne laser altimetry may arise due to uncertainty in the individual datasets, differences in spatial sampling, or changes in ice flow not considered in the firn model. Basin 7.2, for example, contains a number of small marine-terminating glaciers observed to speed up in early summer and slow down in midsummer[76], which may induce corresponding elevation changes not considered by the firn model. Basin 6.2, on the other hand, is a land terminating sector where seasonal elevation changes are likely to be dominated by surface processes.

## Data availability

The interannual elevation change maps, seasonal elevation change and runoff data generated in this study are available for download from Zenodo (https://doi.org/10.5281/zenodo.5562210). The CryoSat-2 data that support this study is available for download from ESA (https://earth.esa.int/web/guest/-/cryosat-products). The IMAU-FDM and RACMOv2p3.2 firn and regional climate model data are available on request from the Institute for Marine and Atmospheric Research Utrecht (imau@science.uu.nl). The MAR regional climate model data are available from (https://www.mar.cnrs.fr/). Operation IceBridge airborne altimetry data are available for download from the National Snow and Ice Data Center (NSIDC) (https://nsidc.org/icebridge/portal/). Ice velocity data are available for download from the NASA Inter-mission Time Series of Land Ice Velocity and Elevation (ITS_LIVE) project (https://its-live.jpl.nasa.gov/#data).

## Code availability

The MATLAB scripts used to process seasonal elevation changes, runoff and plot the main figures are available through Zenodo (https://doi.org/10.5281/zenodo.5503891).

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

## Acknowledgements

This work was supported by NERC through National Capability funding, undertaken by a partnership between the Centre for Polar Observation Modelling and the British Antarctic Survey, and by the European Space Agency's Polar + Earth Observation for Mass Balance study (4000132154/20/I-EF). M.M. was supported by the Lancaster University-UKCEH Centre of Excellence in Environmental Data Science. A.L was supported by the NERC Meltwater Ice-sheet Interactions and the changing climate of Greenland research grant (MII Greenland NE/S011390/1). B.N. was funded by NWO VENI grant VI.Veni.192.019. M.v.d.B and P.K.M. acknowledge support from the Netherlands Earth System Science Centre (NESSC). Computational resources used to perform MAR simulations have been provided by the Consortium des Équipements de Calcul Intensif (CÉCI), funded by the F.R.S. FNRS under grant 2.5020.11 and the Tier-1 supercomputer (Zenobe) of the Fédération Wallonie Bruxelles infrastructure funded by the Walloon Region under grant agreement 1117545. We thank H. Goelzer for providing modelled ice thickness change data used in Supplementary Fig. 5. We acknowledge C. Greene for several MATLAB functions provided in the Climate Data Toolbox, used in visualising the data.

## Author contributions

T.S., A.S., M.M., and K.B. designed the study. T.S., A.S., and M.M. performed the data analysis and wrote the paper. A.L. assisted in the processing and analysis of regional climate model data. L.G. and A.L. assisted in the processing of the altimeter data. P.K.M. carried out the IMAU-FDM simulation, B.N. and M.v.d.B. carried out the RACMO2.3p2 simulation, and X.F. carried out the MARv3.11 simulation. All authors commented on the paper.

## Competing interests

The authors declare no competing interests.
