## [Peer Review File · Nature Communications]

Increased variability in Greenland Ice Sheet runoff from satellite observationsREVIEWER COMMENTS

Reviewer #1 (Remarks to the Author):

Review of "Increased variability in Greenland Ice Sheet runoff from satellite Observations" By Slater et al.

Slater et al. uses satellite altimetry from CryoSat-2 to estimate runoff volume, and in conjunction with SMB and firn densification models, partitions surface mass and elevation changes into seasonal components. This work importantly shows that interannual variability in SMB, and particularly runoff, has increased over several decades as runoff has also increased in total magnitude. The manuscript is well written and concise with overall robust methodological application. Developing improved constraints on runoff quantities outside of heavily utilized regional climate models is pertinent and timely work, it is an overall interesting study, and the manuscript is publishable given the authors address several comments, outlined below. My main critique is on the treatment and underlying assumptions of the dynamic thinning component of surface elevation change, and some parts of the Methods that could be more thoroughly described. Additional clarification/analyses by the authors would help support the argument that runoff is, in fact, being directly measured by satellite altimetry as claimed in the study. As currently presented, I do not think this particular claim can be made given the range of assumptions and uncertainty in the parameters used to address the dynamic component.

Main comments:

As stated above, the claim of providing the first direct observations of runoff from satellite altimetry is not the most accurate framing of the analysis and results presenting here. This claim assumes that dynamic imbalances are minimal or insignificant across the ablation zone, which is not well-supported by the literature (for example, Bevan et al., 2015 (Seasonal dynamic thinning...), Pritchard et al., 2009 (Extensive dynamic thinning on the margins...) and Wood et al., 2021 (Ocean forcing drives glacier retreat in Greenland), etc). I do acknowledge the extra effort made by the authors to correct estimates of runoff-driven elevation change using estimates of steady-state dynamic advection, but several outstanding questions/concerns are described below:

- 1.) How was the 0.5 m threshold, which was used to identify and remove areas of large dynamic imbalances, determined? How sensitive are the resulting RMS values to this threshold? I am concerned that many highly dynamic regions are excluded, and as a consequence, biases estimates of seasonal thinning from runoff high. For example, the manuscript includes an in-text citation to Mouginit et al. 2015 (dynamic-thinning at Zachariæ Isstrøm) when discussing regions of long-term dynamic imbalances, but this glacier is not included in the dynamically imbalanced regions in Figure 2. Given that the relative magnitude of the dynamic component to net elevation change varies on a continuum, I understand the need for a threshold at some point, but it would be helpful to provide more support of the given threshold value. How do overall RMS values vary as a greater proportion of the ice sheet is qualified as dynamically imbalanced and excluded? I suggest adding several references here and discuss potential end member scenarios in highly dynamic regions (for example, something similar to "we estimate that over most of the ablation zone, dynamic thinning is minimal at $<X$ cm/yr, but may reach up to X m/yr, or X during the melt season, in confined areas of rapid glacier change, as shown in X et al and X et al").
- 2.) In Extended Figure 4, the seasonal amplitude observed by CryoSat-2 appears to exceed the modeled amplitude in Basin 7.2, but the amplitudes are more similar in Basin 6.2 (southwest Greenland). Could this be related to the large number of fast flowing outlet glaciers in 7.2 (compared to largely land-terminating 6.2), and an enhanced seasonal dynamic thinning component?

I had difficulty following the Methods section "Computing runoff from CryoSat-2 measurements". One

to two clarifying sentences would improve this section, especially because it is so pertinent to the manuscript. My understanding of this text is that total elevation change from ice advection over the steady state period (1960-1980) is calculated by assuming it is equal to SMB, as provided by the regional climate model. From there, advection constrained to the runoff season is derived from the RCM outputs, and this value is scaled by time and applied to correct for contemporary advection, solving for the true runoff-driven elevation change. However, advection required to maintain steady state during 1960-1980 would be much different than at present, given significant trends in SMB through time. This change is exemplified in Extended Figure 6, where the 1960-1980 derived advection correction is negative in sectors of the contemporary ablation zone. Using extended Figure 5, a negative advection correction implies the drawdown of thinner upstream ice, and thus the correction reduces observed thinning from runoff. This is counterintuitive to the results showing increased runoff observed since 2011. I may be misunderstanding this section, but the use of advection during the historical period as a corrective offset does not seem appropriate in this case.

Minor comments:

Figure 3b

I suggest including the RMS values of Basin 4 in the main text for reference, as well as included as data points in Figure 3b. The authors make a compelling argument as to why this basin and challenging terrain may be problematic for some analyses, but I still think it would be helpful to visualize how relatively anomalous comparisons are within this basin and would provide an upper limit to potential error in problematic topography. Perhaps Basin 4 values could be excluded from the overall fit and RMS statistics with justification, but still included in the figure as a different color or shape for comparison.

In the Methods, section Seasonal elevation changes:

"we find a root mean squared difference of 13 cm between seasonal elevation changes computed from CryoSat-2 where areas of dynamic imbalance are identified (basins 3,5,6,7,8)"

Can you clarify whether this RMS difference is calculated using the full basins that contain dynamic imbalances, or rather the regions of dynamic imbalance themselves?

Additional references to consider:

Moon et al 2020 to line 82 "thinning...as a result of increased surface melting, and the speedup of marine terminating glaciers".

Moon, T. A., Gardner, A. S., Csatho, B., Parmuzin, I., & Fahnestock, M. A. (2020). Rapid reconfiguration of the Greenland Ice Sheet coastal margin. *Journal of Geophysical Research: Earth Surface*, 125, e2020JF005585. <https://doi.org/10.1029/2020JF005585>

King et al 2020 to line 40 "The Greenland ice sheet has lost mass over recent decades because ice discharge has exceeded its SMB"

King, M.D., Howat, I.M., Candela, S.G. et al. Dynamic ice loss from the Greenland Ice Sheet driven by sustained glacier retreat. *Commun Earth Environ* 1, 1 (2020). <https://doi.org/10.1038/s43247-020-0001-2>

Pritchard et al 2009 to Line 174 "While there are many examples of long-term ice dynamic imbalance across the Greenland ice sheet..."

Pritchard, H. D., Arthern, R. J., Vaughan, D. G. and Edwards, L. A.: Extensive dynamic thinning on the margins of the Greenland and Antarctic ice sheets, *Nature*, 461, 971, 2009.

Reviewer #2 (Remarks to the Author):

Review of "Increased variability in Greenland Ice Sheet runoff from satellite observations" by Slater and coauthors (NCOMMS-20-50187-T).

This study utilizes satellite-derived ice sheet elevation measurements from CryoSat-2. Combined with a modeled, climatological steady-state ice sheet surface mass balance and observed density estimates from firn cores, the work purports to estimate surface runoff over the Greenland Ice Sheet for the period 2011-2020. The results are found to be in close agreement with regional climate model estimates, and confirm several aspects of those results. The period examined is marked by considerable interannual variability, with large runoff amounts occurring in 2012 and 2019, years of known significant surface melt.

The main manuscript is approximately 3221 words with 4 figures and 1 table. A supporting information section contains a discussion of methods, along with 7 extended data figures and 1 table. I would recommend minor revision. I have three main points for the authors to consider, followed by additional minor points below.

1. The manuscript provides a means for quantitatively assessing summertime runoff on the ice sheet from a newly developed approach. The authors are careful to examine the intermediate step of determining CryoSat-2 elevation change by comparing with a firn model and with IceBridge field results. But this tends to make the methods somewhat difficult to follow. One might assume that the firn model plays a role in the computation of runoff without a careful reading of the methods section. The authors may wish to consider a re-ordering of the methods section, so that "Computing runoff from CryoSat-2 measurements" immediately follows the first segment, "Computing elevation and elevation change from CryoSat-2 measurements". While the current ordering follows the main text, the suggested re-ordering would more concisely indicate all the steps leading to the runoff variable.

2. As this is in fact an effort towards a "direct measurement" of runoff that is largely absent of surface modeling, it is a bit unclear as to how other surface terms including summertime snowfall, sublimation, and windblown divergence are addressed. This could be more clearly indicated in the text.

3. Essentially, an observation of (a.) the total surface elevation change is paired with (b.) an estimate of elevation change due to ice divergence, yielding a residual term associated with surface mass ablation (lines 527-538). There have been other recent efforts to provide a comparable combination (e.g., Smith et al. 2020 AGU meeting abstract C034-06), but this manuscript is unique in providing a plausible, long-term record of derived surface runoff from altimetry. The selection of a modeled, steady-state ice sheet surface mass balance as the second component here is rather intriguing. SAR velocities are used in the manuscript for computing the total surface height change in a small area of the ice sheet (line 446), why wouldn't they be used in estimating runoff? For example, Fettweis et al. (2020, ref. #20) employed SAR-derived discharge estimates in combination with gravimetric observations to yield surface mass balance for evaluating regional climate models.

Additional points

- Line 88: "airborne"
- Line 435: suggest inserting comma after "not"
- Line 444: "change"
- Line 537: z_f is first indicated here, how is it computed?
- Line 544: "We also identify and exclude areas of long-term dynamical imbalance.." Can an estimate of this area be provided.

- Figure 1 contains a very faint lat/lon grid that could probably be done away with. Otherwise the figures are clear and of good quality.

We thank the reviewers for their positive and constructive reviews of our manuscript, and we have revised our manuscript to address their concerns. These changes have substantially improved our manuscript. In summary, the main changes are:

- We have improved our scheme to classify areas of ice dynamical imbalance by incorporating a time series of ice velocity measurements, as suggested by Reviewer #1. Although this scheme identifies new areas of dynamical imbalance, it has a small (1 %) impact on our estimated runoff.
- We have significantly revised and improved our description of the correction for steady-state divergence, as requested by both Reviewers. This includes using ice-sheet model simulations to support our use of the long-term SMB mean as a proxy for steady-state divergence.
- We have re-structured the methods section for readability, as requested by Reviewer #2.
- We have added an improved description of our treatment of other surface processes, and their effects on our runoff solution, as requested by both Reviewers. This includes an assessment of their magnitude and the steps we take to mitigate their impact.

We have also made numerous other changes and additions to the text and figures, following the suggestions of both Reviewers. Our response to each of the comments raised is included below; Reviewer comments are in *black italic* and our responses are in blue. We have included a tracked changes version of the manuscript after our responses, so it is clear where changes have been made.

Reviewer #1

Slater et al. uses satellite altimetry from CryoSat-2 to estimate runoff volume, and in conjunction with SMB and firn densification models, partitions surface mass and elevation changes into seasonal components. This work importantly shows that interannual variability in SMB, and particularly runoff, has increased over several decades as runoff has also increased in total magnitude. The manuscript is well written and concise with overall robust methodological application. Developing improved constraints on runoff quantities outside of heavily utilized regional climate models is pertinent and timely work, it is an overall interesting study, and the manuscript is publishable given the authors address several comments, outlined below. My main critique is on the treatment and underlying assumptions of the dynamic thinning component of surface elevation change, and some parts of the Methods that could be more thoroughly described. Additional clarification/analyses by the authors

would help support the argument that runoff is, in fact, being directly measured by satellite altimetry as claimed in the study. As currently presented, I do not think this particular claim can be made given the range of assumptions and uncertainty in the parameters used to address the dynamic component.

Thank you for your careful review of our manuscript. We have improved our treatment of dynamic areas and descriptions of how our chosen assumptions affect our runoff solution (please see detailed responses below), which we hope addresses your concerns.

As stated above, the claim of providing the first direct observations of runoff from satellite altimetry is not the most accurate framing of the analysis and results presenting here. This claim assumes that dynamic imbalances are minimal or insignificant across the ablation zone, which is not well-supported by the literature (for example, Bevan et al., 2015 (Seasonal dynamic thinning...), Pritchard et al., 2009 (Extensive dynamic thinning on the margins...) and Wood et al., 2021 (Ocean forcing drives glacier retreat in Greenland), etc). I do acknowledge the extra effort made by the authors to correct estimates of runoff-driven elevation change using estimates of steady-state dynamic advection, but several outstanding questions/concerns are described below:

1.) How was the 0.5 m threshold, which was used to identify and remove areas of large dynamic imbalances, determined? How sensitive are the resulting RMS values to this threshold? I am concerned that many highly dynamic regions are excluded, and as a consequence, biases estimates of seasonal thinning from runoff high. For example, the manuscript includes an in-text citation to Mouginot et al. 2015 (dynamic-thinning at Zachariæ Isstrøm) when discussing regions of long-term dynamic imbalances, but this glacier is not included in the dynamically imbalanced regions in Figure 2. Given that the relative magnitude of the dynamic component to net elevation change varies on a continuum, I understand the need for a threshold at some point, but it would be helpful to provide more support of the given threshold value. How do overall RMS values vary as a greater proportion of the ice sheet is qualified as dynamically imbalanced and excluded?

Thank you for raising this - we agree that our original classification of dynamic areas lacked justification in the text, and omitted key glaciers, as you have rightly pointed out.

To address this, we have defined a new dynamic mask, based upon the trend and variability in velocity time series acquired using optical imagery between 1985-2018 (Extended Data Figure 4). Because these data don't provide complete spatial coverage over the ice sheet, we use these velocity data to determine a dynamic elevation change threshold, which we find to be 0.4 m/yr, and apply across the remainder of the ice sheet. As well as being based upon independent data, this new mask now includes several key glaciers which were omitted: Zachariæ Isstrøm, Petermann, and Helheim, which exhibits seasonal variability as described in Bevan et al. 2015. We note that, as Bevan et al. report, seasonal changes in elevation due to changes in ice flow are confined to, and are largest in amplitude, in the fastest flowing regions at low elevations near the glacier terminus, which are now accounted for in our updated mask. Beyond these dynamic regions, and in particular during the summer, surface elevation changes are dominated by SMB fluctuations.

We have also performed a sensitivity test on the choice of this dynamic elevation threshold, using a series of thresholds (Extended Data Table 1). We find our runoff solution is relatively insensitive to the rate selected to mark areas of dynamical imbalance, varying by only 36 Gt/yr for rates between 0.2 m/yr and 1.0 m/yr.

To make this clearer to the reader, we have added (1) clarifying text to the main text, citing the papers you have mentioned here that we had omitted previously and (2) a detailed description of the methods used to generate the mask and the threshold sensitivity analysis, including 1 figure (Extended Data Figure 4) and 1 table (Extended Data Table 1) to illustrate the data used.

I suggest adding several references here and discuss potential end member scenarios in highly dynamic regions (for example, something similar to “we estimate that over most of the ablation zone, dynamic thinning is minimal at <X cm/yr, but may reach up to X m/yr, or X during the melt season, in confined areas of rapid glacier change, as shown in X et al and X et al”).

Thanks for this suggestion – we have added a sentence to this effect in the main text.

2.) In Extended Figure 4, the seasonal amplitude observed by CryoSat-2 appears to exceed the modeled amplitude in Basin 7.2, but the amplitudes are more similar in Basin 6.2 (southwest

Greenland). Could this be related to the large number of fast flowing outlet glaciers in 7.2 (compared to largely land-terminating 6.2), and an enhanced seasonal dynamic thinning component?

Thanks for raising this - while this may be the case, there are several compounding factors which make precisely determining reasons for differences between these datasets difficult. Along with the possibility of enhanced seasonal dynamic thinning in these regions, there are uncertainties within the individual datasets which may also cause the reported differences. Although we did dedicate a paragraph to discussing the reasons and physical processes which may cause these differences in the main text (see Seasonal cycle of melting and snowfall), we appreciate this was not clear in the methods text where the figure appears. To address this, we have added some text to the methods explaining the difference between the geography of these regions, and why differences between these datasets may arise.

On the ice sheet scale, we demonstrate that seasonal elevation changes in elevation recorded by CryoSat-2 and IMAU-FDM agree very well and are highly correlated, which reinforces our approach to use CryoSat-2 to estimate ice sheet-wide runoff.

I had difficulty following the Methods section "Computing runoff from CryoSat-2 measurements". One to two clarifying sentences would improve this section, especially because it is so pertinent to the manuscript. My understanding of this text is that total elevation change from ice advection over the steady state period (1960-1980) is calculated by assuming it is equal to SMB, as provided by the regional climate model. From there, advection constrained to the runoff season is derived from the RCM outputs, and this value is scaled by time and applied to correct for contemporary advection, solving for the true runoff-driven elevation change. However, advection required to maintain steady state during 1960-1980 would be much different than at present, given significant trends in SMB through time. This change is exemplified in Extended Figure 6, where the 1960-1980 derived advection correction is negative in sectors of the contemporary ablation zone. Using extended Figure 5, a negative advection correction implies the drawdown of thinner upstream ice, and thus the correction reduces observed thinning from runoff. This is counterintuitive to the results showing increased runoff observed since 2011. I may be misunderstanding this section, but the use of advection during the historical period as a corrective offset does not seem appropriate in this case.

Again, thanks for raising this. We agree with you that our description of our assumptions regarding the correction for steady-state ice divergence were not clear. Within the runoff zone, and away from identified areas of dynamical imbalance, we assume that because the dynamic response to changes in SMB occurs over multi-decadal timescales, thickness changes due to ice divergence are approximately equal to the long term mean, and therefore present-day observed departures in thickness are dominated by changes in SMB. To demonstrate the validity of our assumption, and in the absence of direct observations, we compare dynamic and SMB-related ice thickness changes derived from an ensemble of ISMIP6 model experiments, taken from Goelzer et al 2020. From these projections we find that, even at 2100 when the ice sheet has been out of balance for ~100 years and there is widespread dynamical imbalance, elevation changes due to SMB (-0.86m/yr) in the region we perform our runoff calculation are almost 30 times higher than those due to ice dynamics (-0.03 m/yr). Because the ice sheet was approximately in balance pre-2000s, this suggests that contemporary changes in ice thickness reflect SMB anomalies, not divergence anomalies, and our choice of using the long-term SMB mean as a proxy for steady-state ice divergence is valid.

We have re-written this entire section of the methods within the main paper, and added some clarifying sentences, along with a new figure (Extended Data Figure 5) and a new table (Extended Data Table 2) to better explain and justify our assumptions.

I suggest including the RMS values of Basin 4 in the main text for reference, as well as included as data points in Figure 3b. The authors make a compelling argument as to why this basin and challenging terrain may be problematic for some analyses, but I still think it would be helpful to visualize how relatively anomalous comparisons are within this basin and would provide an upper limit to potential error in problematic topography. Perhaps Basin 4 values could be excluded from the overall fit and RMS statistics with justification, but still included in the figure as a different color or shape for comparison.

Thank you for this suggestion - we have added the data points for Basin 4 in Figure 3, and added details of the RMS differences with this basin included in the main text.

“we find a root mean squared difference of 13 cm between seasonal elevation changes computed from CryoSat-2 where areas of dynamic imbalance are identified (basins 3,5,6,7,8)”

Can you clarify whether this RMS difference is calculated using the full basins that contain dynamic imbalances, or rather the regions of dynamic imbalance themselves?

Apologies our wording wasn't clear on this - we meant the full basins which contain dynamic imbalance, and have edited the text to make this clear.

Additional references to consider:

Moon et al 2020 to line 82 "thinning...as a result of increased surface melting, and the speedup of marine terminating glaciers".

*Moon, T. A., Gardner, A. S., Csatho, B., Parmuzin, I., & Fahnestock, M. A. (2020). Rapid reconfiguration of the Greenland Ice Sheet coastal margin. *Journal of Geophysical Research: Earth Surface*, 125, e2020JF005585. <https://doi.org/10.1029/2020JF005585>*

King et al 2020 to line 40 "The Greenland ice sheet has lost mass over recent decades because ice discharge has exceeded its SMB"

*King, M.D., Howat, I.M., Candela, S.G. et al. Dynamic ice loss from the Greenland Ice Sheet driven by sustained glacier retreat. *Commun Earth Environ* 1, 1 (2020). <https://doi.org/10.1038/s43247-020-0001-2>*

Pritchard et al 2009 to Line 174 "While there are many examples of long-term ice dynamic imbalance across the Greenland ice sheet..."

*Pritchard, H. D., Arthern, R. J., Vaughan, D. G. and Edwards, L. A.: Extensive dynamic thinning on the margins of the Greenland and Antarctic ice sheets, *Nature*, 461, 971, 2009.*

Thank you for mentioning these papers - we initially submitted the article to Nature Geoscience, which had a reference limit of 50, and it was automatically transferred to *Nature Communications*. As this limit is relaxed for Nature Communications, we are more than happy to add these references to the text.

Reviewer #2

This study utilizes satellite-derived ice sheet elevation measurements from CryoSat-2. Combined with a modeled, climatological steady-state ice sheet surface mass balance and observed density estimates from firn cores, the work purports to estimate surface runoff over the Greenland Ice Sheet for the period 2011-2020. The results are found to be in close agreement with regional climate model estimates, and confirm several aspects of those results. The period examined is marked by considerable interannual variability, with large runoff amounts occurring in 2012 and 2019, years of known significant surface melt.

The main manuscript is approximately 3221 words with 4 figures and 1 table. A supporting information section contains a discussion of methods, along with 7 extended data figures and 1 table. I would recommend minor revision. I have three main points for the authors to consider, followed by additional minor points below.

Thank you for your careful review of our manuscript – please find detailed responses to your points below.

1. The manuscript provides a means for quantitatively assessing summertime runoff on the ice sheet from a newly developed approach. The authors are careful to examine the intermediate step of determining CryoSat-2 elevation change by comparing with a firn model and with IceBridge field results. But this tends to make the methods somewhat difficult to follow. One might assume that the firn model plays a role in the computation of runoff without a careful reading of the methods section. The authors may wish to consider a re-ordering of the methods section, so that “Computing runoff from CryoSat-2 measurements” immediately follows the first segment, “Computing elevation and elevation change from CryoSat-2 measurements”. While the current ordering follows the main text, the suggested re-ordering would more concisely indicate all the steps leading to the runoff variable.

Thank you for this suggestion - we have moved the section “Computing runoff from CryoSat-2 measurements” further up in the methods. However, instead of putting it immediately after “Computing elevation and elevation change from CryoSat-2 measurements”, we have placed it after “Seasonal elevation changes”, as we feel it is beneficial for the reader to know about the computation of seasonal elevation changes prior to runoff (as they form the basis of the runoff calculation). This way all the methods relating to the processing of CryoSat-2 data are still grouped together, which we hope still avoids any potential confusion.

2. As this is in fact an effort towards a “direct measurement” of runoff that is largely absent of surface modeling, it is a bit unclear as to how other surface terms including summertime snowfall, sublimation, and windblown divergence are addressed. This could be more clearly indicated in the text.

Thank you for raising this. Indeed our approach is a simplification because it neglects mass accumulated during the summer period. In the runoff zone, and outside areas of dynamical imbalance, we assume that (1) observed summertime elevation changes represent the surface height changes due to the integral of all SMB processes and (2) during our defined summer period, the SMB is dominated by runoff. To test the validity of these assumptions, we compare modelled summer SMB and annual runoff within the ablation and inland runoff zones, and find them to be correlated ($R^2 = 0.99$ and 0.85 , respectively), which indicates that the neglected processes play a minor role.

However, as you have correctly pointed out, we have not described the effects of other surface processes on our solution. Although our method does not account for sublimation and drifting snow erosion, these represent small contributions to the overall SMB (10 Gt/yr and less than 1 Gt/yr across the entire ice sheet, respectively). During our observational period, summertime snowfall is also small (22 Gt and 32 Gt within the ablation and inland runoff zones, respectively) and amounts to 13% of the total runoff, on average.

The effects of snowfall on our solution will depend on the conditions within a given year, and can be broadly split into two scenarios:

1. The melting of a thick surface snow layer at the start of the runoff season, present due to heavy snowfall in the preceding winter for example, could bias our runoff estimates

high due to our choice of higher densities in both the ablation and inland runoff zones. We note that in the inland runoff zone, where snowfall is higher, our choice of a lower density acts to reduce this bias in part, and that modelled surface densities typically vary by 14 kg/m^3 and 9 kg/m^3 at the beginning of May in the ablation and inland runoff zones, respectively.

2. Due to our 60-day temporal sampling, our solution will not capture snowfall in summer and underestimate the total runoff. However, although this simplification could in principle lead to an underestimate of runoff, our choice of ice and firn densities mitigate this because they represent upper and contemporary bounds, respectively, and in any case the simplification has only a minor impact because snowfall is a small fraction (15 %) of the summertime SMB across the runoff zone.

Taking both of these effects into account, snowfall equal to the modelled average will bias our runoff by 5 % and 21 % within the ablation and inland runoff zones, respectively, and 8 % overall.

We have added the above discussion to both the main text and the methods to make our treatment of other surface processes clear to the reader. Despite these simplifications, we demonstrate that our approach is able to reliably capture the magnitude and variability of Greenland runoff, and we hope that we can explicitly correct for the effects of snowfall in future.

3. Essentially, an observation of (a.) the total surface elevation change is paired with (b.) an estimate of elevation change due to ice divergence, yielding a residual term associated with surface mass ablation (lines 527-538). There have been other recent efforts to provide a comparable combination (e.g., Smith et al. 2020 AGU meeting abstract C034-06), but this manuscript is unique in providing a plausible, long-term record of derived surface runoff from altimetry. The selection of a modeled, steady-state ice sheet surface mass balance as the second component here is rather intriguing. SAR velocities are used in the manuscript for computing the total surface height change in a small area of the ice sheet (line 446), why wouldn't they be used in estimating runoff? For example, Fettweis et al. (2020, ref. #20) employed SAR-derived discharge estimates in combination with gravimetric observations to yield surface mass balance for evaluating regional climate models.

Thank you - a similar point was also raised by Reviewer #1. Because GRACE gives the total mass change, it is possible to estimate SMB related mass changes directly by differencing discharge estimated from velocity data, as in Fettweis et al., 2020. Our situation is different, because the altimetry reflects the deviation in mass change relative to the steady state. Therefore, to isolate the SMB related change, we need to correct for the height change associated with the steady state divergence. Unfortunately, it is not possible to estimate this from velocity data because observations from when the ice sheet was in a steady-state are sparse in both space and time. Instead, we choose to use the modelled long-term SMB mean as a proxy for the steady state divergence. We appreciate, however, that we did not adequately describe or justify this approach in the submitted version. As requested by Reviewer #1 (please see our response above), we have added both text and figures to the manuscript to better describe and justify our correction for steady-state divergence

Line 88: "airborne"

Fixed, thanks.

Line 435: suggest inserting comma after "not"

Done, thanks.

Line 444: "change"

Fixed, thanks.

Line 537: zf is first indicated here, how is it computed?

Beyond areas excluded by our ice dynamical mask, z_f is assumed to be the total summer elevation change recorded by CroSat-2. We have added text to this section in order to make this clearer to the reader.

Line 544: "We also identify and exclude areas of long-term dynamical imbalance.." Can an estimate of this area be provided.

Apologies - we had included an estimate of the area in the main text, but neglected to include it here. As suggested by Reviewer #1, we have significantly revised our definition of areas of dynamical imbalance to include areas identified by independent ice velocity data. As a result, we have added new text to the methods to describe this, and included an estimate of the area as requested here.

Figure 1 contains a very faint lat/lon grid that could probably be done away with. Otherwise the figures are clear and of good quality.

Thanks, we have removed the lat lon grid.

REVIEWERS' COMMENTS

Reviewer #1 (Remarks to the Author):

Slater et al. 2021 have carefully and fully addressed my comments on the initial manuscript. The inclusion of Extended data table 1 and 2, and the reworking on the section on dynamic region masking (with an extra sensitivity analysis for 0.2m -1.0 m masking thresholds) greatly improve the paper. I think the paper in its current form is of high quality and the analysis rigorous, therefore meeting the standards of publication in Nature Communications. The revised manuscript was a pleasure to read.

Reviewer #1 (Remarks to the Author):

Slater et al. 2021 have carefully and fully addressed my comments on the initial manuscript. The inclusion of Extended data table 1 and 2, and the reworking on the section on dynamic region masking (with an extra sensitivity analysis for 0.2m -1.0 m masking thresholds) greatly improve the paper. I think the paper in its current form is of high quality and the analysis rigorous, therefore meeting the standards of publication in Nature Communications. The revised manuscript was a pleasure to read.

Thank you Reviewer #1 for taking the time to review the revised version of our manuscript, we are very pleased that we have been able to address your concerns.